# Person-centred care by a combined digital platform and structured telephone support for people with chronic obstructive pulmonary disease and/or chronic heart failure: study protocol for the PROTECT randomised controlled trial

Lilas Ali [ORCID],[1,2,3] Sara Wallström [ORCID],[1,2] Emmelie Barenfeld [ORCID],[1,2] Andreas Fors,[1,2,4] Eva Fredholm,[5] Hanna Gyllensten [ORCID],[1,2] Karl Swedberg,[1,2,6,7] Inger Ekman[1,2,6]

**Correspondence to**
Dr Lilas Ali; lilas.ali@gu.se

## ABSTRACT

**Background** A core feature of chronic obstructive pulmonary disorder (COPD) and chronic heart failure (CHF) is that symptoms may change rapidly because of illness progression. Thus, these chronic conditions are associated with high rehospitalisation rates. Person-centred care (PCC) has been shown to have several benefits for patients with COPD or CHF (or both disorders) but it has not yet been investigated through e-health services.

**Aim** The project aims to evaluate the effects of PCC by a combined digital platform and structured telephone support for people with COPD and/or CHF.

**Methods and analysis** A randomised controlled trial with open, parallel groups which employs a participatory design process will be used. This project will also include process and health economic evaluation of the intervention.

**Ethics and dissemination** Ethical approval has been secured from the Regional Ethical Review Board in Gothenburg, Sweden (Dnr 063-17 and T063-18). Results will be presented at conferences and to healthcare professionals, participants and patient organisations. Findings will also be submitted for publication in peer-reviewed journals.

**Trial registration number** NCT03183817

## Strengths and limitations of this study

► This project is the first randomised controlled trial to evaluate the effects of person-centred care by a combined digital platform and structured telephone support for people with chronic obstructive pulmonary disorder and/or chronic heart failure.

► The protocol details a complex intervention aimed at operationalising person-centred ethics.

► To answer the research questions of the study, collected data will be analysed using quantitative and qualitative methods.

► The evaluation will include patient-reported outcome measures together with health economics and process outcomes.

► Unfortunately, this study did not include individuals without a device for the internet and those who could not read or write in Swedish.

## INTRODUCTION

Chronic heart failure (CHF) and chronic obstructive pulmonary disease (COPD), two long-term conditions common in older populations, are associated with high rehospitalisation rates and high mortality.[1][2] In addition, both conditions severely affect activities of daily living because of frequent symptoms that limit health-related quality of life (HRQoL), compelling patients to be dependent on others.[3] Patients with CHF and/or COPD frequently require hospital care and the severity of their symptoms often leads to repeated visits to hospital, outpatient or primary care.[1–3] Although pharmacological therapy has improved outcomes markedly over the past 10–15 years, management programmes are still needed to optimise care.[4] There are benefits of pulmonary rehabilitation (eg, reduction in readmission and mortality), including exercise training, education and self-management interventions that specifically target behaviour change for patients with COPD.[5] A consistent recommendation in the treatment of CHF is management programmes designed to improve outcomes through structured follow-ups that increase patients' self-care skills and other key issues (eg, symptoms and

disease trajectory).[2] Being affected by a chronic disease with a high symptom burden is a major challenge to living a full life. Healthcare professionals (HCPs), such as registered nurses, occupational therapist and physiotherapist, in conjunction with well-developed technology, can serve to enable successful self-management for individuals with severe long-term illnesses.[6]

## Person-centred care

The basis of person-centred care (PCC) is to include the patient's experience, situation, capabilities and resources in the care process. PCC highlights the importance of understanding the person behind the patient as a human being with reason, will, emotions and needs in order for that person to be able to be an active partner in his or her care and treatment.[7] Patients (often with the help of relatives) present themselves as persons by expressing their illness narrative. This narrative includes how daily life events are being affected by the condition and treatment and creates a starting point for collaborative care and partnership between HCPs and patients which encourages and empowers patients to actively take part in finding solutions to their problems.[7 8] PCC entails not only to identify health barriers but also to confirm human capabilities and opportunities in the patients' home and local environment. PCC is based on ethical principles, inspired by the capability approach (an alternative approach to welfare economics), which has also been used in research as a theoretical frame of reference in several disciplines (eg, in economics by the Nobel Laureate Amartya Sen).[9] All people develop their capabilities (such as a sense of responsibility for oneself and others) in relation to other people.

Previous studies evaluating PCC that, for example, uses personal health plans and symptom tracking have shown positive effects in patients with several conditions, both acute and long term, and in different care settings. Positive outcomes include shortened hospital stays,[10] reduced readmittance,[11] improved self-efficacy,[12] reduced healthcare costs,[13 14] maintained functional performance,[10] improved HRQoL,[11] improved pain management, greater stamina and enhanced self-reported health,[15] increased physical activity,[16] improved self-efficacy to manage symptoms,[17] reduced fatigue and anxiety[15] and improved experience of the care.[18] Moreover, effects have been especially prominent in patients without postsecondary education,[19] an effect that was sustained for up to 2 years.[20] These studies have applied a PCC approach in practice, wherein the needs of the patients are important which also incorporate their human ability, which is an often underused resource in healthcare.[7 21 22]

## E-health

To meet demands on accessible and efficient care, one suggestion is to facilitate the partnership between HCPs and patients.[23] One such effort can be the care and support provided through e-health. At present, the internet is frequently used as a source of information

and Sweden belongs to the countries that is high up in the development of maturity of internet use concerning the number of computers and the possibility of accessing the internet.[24] Digital web-based care has no geographical constraints, that is, it is potentially accessible everywhere. It also affords the possibility to reach persons who at present avoids to seek help because they find it stigmatising.[25] Moreover, the possibility of having contact with HCPs in privacy, when one has the time and when questions arise, is found to be important.[26] Many people want to manage their health by themselves, especially when they are concerned about their illness. Yet people yearn to be understood and they communicate with others with similar experiences.[26] E-health solutions offer these possibilities. Digital health interventions have also been shown to hold promise to improve the quality and experience of care for patients with CHF.[27]

There are, however, some concerns that need to be taken into account when designing e-health solutions. There are some indications that older people can find e-health solutions difficult to handle.[28] Previous research have also shown that e-health support, in which the users are not directly involved in the process, has low impact.[29]

There is some research combining e-health with PCC. PCC in the form of a structured telephone support for patients with CHF and/or COPD has been shown to mitigate worsening self-efficacy without increasing the risk of clinical events.[30] Using a digital symptom tracking tool in combination with PCC has been found to improve general self-efficacy compared with standard care.[31] E-health interventions that are developed using a participatory design and that use personal resources are also consistent with the ethics expectations of PCC. Overall, e-health solutions are considered a promising approach to strengthen self-management among patients with CHF or COPD.[29] There is, however, a need for studies investigating the effects of PCC via e-health solutions for people with long-term illness.

## AIM

The project aims to evaluate the effects of PCC by a combined digital platform and structured telephone support for people with COPD and/or CHF.

## METHODS/DESIGN

The design will be a randomised controlled trial (RCT) with open parallel groups. In addition, the project will include process and health economic evaluations of the study. Thus, the project will integrate the collection and analysis of both qualitative and quantitative data to answer the research questions. User experiences and process evaluations are necessary complements to each other to help understand more about enablers and barriers to self-management support using a combined digital platform and telephone support. The protocol complies with the

Standard Protocol Items: Recommendations for Interventional Trials.[32] [33]

## Patient and public involvement

Public involvement is a cornerstone in PCC. Therefore, this study incorporates a participatory design that assumes that all users (patients, relatives and HCPs) are involved in the study design, which has been reported to facilitate implementation.[34] Extensive consultation with members of the community took place during the development of the digital platform and design of the study. During these consultations, the patient representatives requested that the platform should include functionalities such as ability for two-way communication, reliable information and advice about their condition and that it should be user friendly. The HCPs contributed with input concerning layout and technical functionalities of the platform. An advisory group consisting of patient representatives, relatives, HCPs, system developers and researchers was formed to provide advice and codesign all of the major elements of the study design.

## Participants and setting

Study participants who fulfil the inclusion criteria of the study will be recruited from nine public primary care centres in Gothenburg.

### Inclusion criteria

► Diagnosed with COPD or CHF.
► Listed at one of the nine participating primary care centres.
► Must understand written and spoken Swedish.

### Exclusion criteria

► No device with internet access, that is, no computer, smartphone or tablet.
► Severe impairment that prevents persons from using e-health support.
► No registered address.
► Any severe disease with an expected survival of <12 months.
► Cognitive impairment.
► Ongoing documented diagnosis of alcohol/drug abuse.
► Other diseases that can interfere with follow-up (eg, severe depression and other severe mental illnesses).
► Participating in another conflicting study.

### Enrolment and randomisation

Designated HCPs will screen medical records of listed patients at the participating primary care centres against study inclusion and exclusion criteria. Eligible patients will be sent a letter informing them about the study. The HCPs then contact the eligible patients through phone to give further information about the study and ask whether they are willing to participate. If the patient agrees to participate, a consent form together with a return prepaid envelope will be sent to the patient. Once the consent form is returned to the HCPs, patients will be randomised to usual care or PCC plus usual care. The randomisation will be based on a computer-generated list created by a third party. All patients will be informed about group assignments through phone.

## Usual care

Patients randomised to usual care will be managed by regular evidence-based treatment and care as outlined in treatment guidelines.[2] [5]

## Design of intervention

The intervention is based on the theoretical PCC framework in which all people are seen as capable with a unique understanding of themselves and who have unique experiences, expectations, needs, preferences and resources. A central component of PCC is that the professional and patient jointly develop a personal health plan using resources identified in each patient's illness history as well as defining potential barriers to effective care.[7] Self-efficacy as a concept is closely related to PCC as it also refers to people's belief in their capability to affect daily life events.[7] [31] [35] In addition, self-efficacy has been shown to have important implications for patient recovery,[36] which is related to the fact that patients confident in their ability to manage everyday life are likely to engage in rehabilitation more actively.[7] [35] Steps and components of the intervention are described as follows.

### Description of intervention elements

When the HCPs call the participants in the intervention group to inform them about the outcome of the randomisation procedure, the HCPs will also describe the intervention, assist the participants in creating a login to the platform and describe its features and decide on a date and time for a phone call when the healthcare plan is created. All the dedicated HCPs participated in the development of the digital platform and received special training in performing PCC communication at a distance. Access to the digital platform will be password protected.

### Telephone support and health plan

When the HCPs call the participants, they listen to their narrative about daily life events and how they are affected by the condition, what kind of problems they experience and what resources they have to cocreate the health plan. The HCPs try to encourage narration and establish a partnership by using PCC communication skills such as open-ended questions, reflections and summaries. Based on the patient narrative, patient goals, resources and needs are identified and communicated with the patient (participant). The participant (sometimes together with relatives) and the HCPs formulate a person-centred health plan in partnership. The health plan contains five elements:

1. Today we have talked about.
2. You would like to be able to do/feel.
3. To get there you have to.
4. Capabilities and resources that could help you are.
5. Support that you might need is.

The health plan will be based on mutual agreement between the patient and HCPs and contains the patient's goals, resources and support needs. It also contains follow-up actions and steps needed to achieve progress in the care of the patient. The date of follow-up conversations will be scheduled jointly. This plan will be uploaded to the digital platform with help from the HCPs if the participants have chosen to write the plan themselves or it will be written directly by the HCPs and then finalised when accepted by the participants.

The health plan will be the point of departure for the forthcoming conversations and communication via the platform that the participants and the HCPs will have during the study period (of 6 months). The plan will be considered during each follow-up when the HCPs contact the participants and revised when necessary. During the intervention, the participants will also be free to contact the HCPs during office hours.

### Digital platform

The platform will contain headings, such as messages, daily ratings, health plan, contact, next of kin and links, when logged in that may inspire the patients to make notes on 'a good day' and 'a bad day', respectively. Functionalities of the platform include ability for two-way communication, possibility to rate symptoms and a collection of links to information concerning COPD and CHF. Symptom ratings and comments will also be made to create a graph over time once the daily ratings have been completed. The HCPs will be able to see the patients' accounts and make comments. The patients will be able to add or delete any person (eg, informal carers, family or friends) to whom they would like to give access to their account. The patients will also be able to customise what content they want visible for invited persons. For instance, they could choose to restrict visibility of the messages between themselves and the HCPs or elect to show all contents in their account such as the health plan and daily ratings.

The digital platform will also contain a function to send private messages between the participants and the HCPs. The HCPs check for new messages daily. Another function offered in the platform will be the possibility to link relevant information about CHF and COPD which is provided by patient organisations (eg, the heart and lung association) and the Swedish national help guide ( 1177.se). There will also be links to a peer-to-peer support group.

### Sources and collection of data

Data will be collected using self-report instruments, interviews, study documentation, medical records and administrative registers. Questionnaires will be mailed (together with a prepaid return envelope) to the participants at study inclusion and 3, 6, 12 and 24 months after randomisation. Paper questionnaires were chosen since not all of the instruments are validated for online use. Follow-ups and adverse events (ie, admission to hospital or death) will be controlled for all participants. Baseline data (including demographics, characteristics and medical history) will be collected from the patients' medical records. For details, see figure 1.

### Questionnaires

► The General Self-Efficacy Scale.[37]
► EuroQol-5 Dimensions health state questionnaire (EQ-5D).[38]
► Hospital Anxiety and Depression Scale (HADS).[39]
► Shortness of Breath in Heart Failure (SOB-HF).[40]
► COPD Assessment Test (CAT).[41]
► The Medical Research Council (MRC) breathlessness scale.[42]

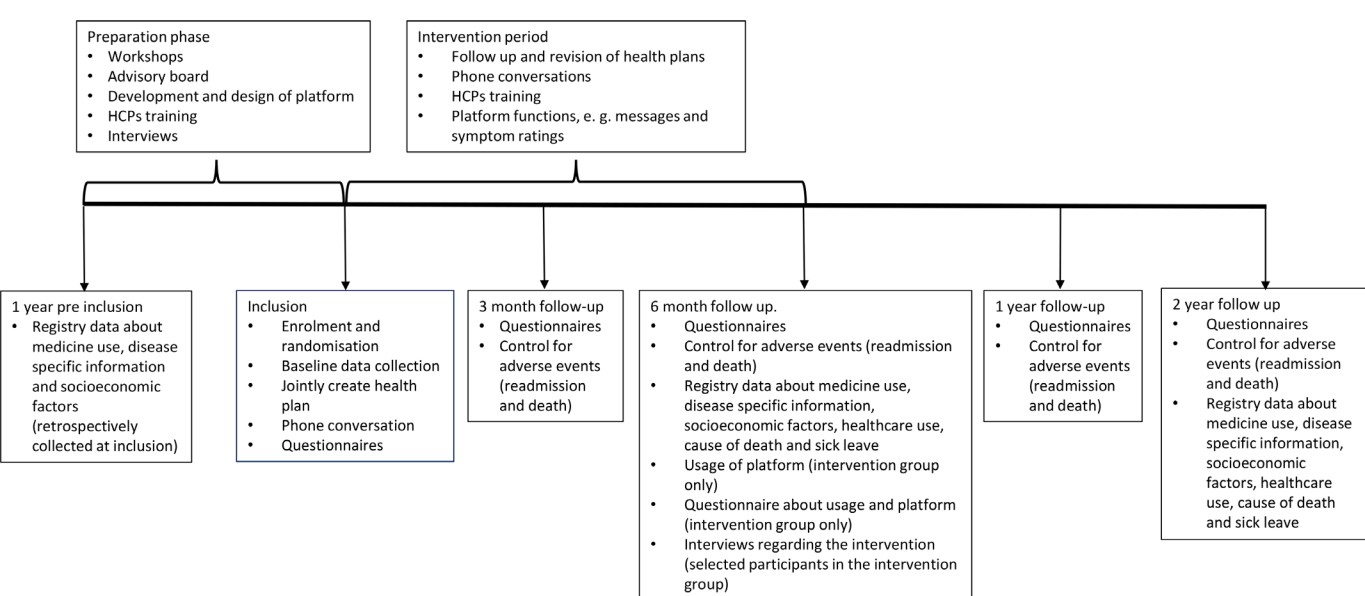

**Figure 1** Timeline of the intervention and data collection. HCPs, healthcare professionals.

## Primary end point

The primary end point will be the composite score of change in general self-efficacy and rehospitalisation or death 6 months after randomisation.

A patient will be classified as deteriorated if:

► General self-efficacy has decreased by ≥5 units.
► Patient has been hospitalised for unscheduled reasons due to COPD/CHF.
► Patient has died due to all causes.

A patient is classified as improved if:

► General self-efficacy has increased by ≥5 units.
► Patient has not been hospitalised due to COPD/CHF.
► Patient has not died.

Those who have neither deteriorated nor improved are considered unchanged.

## Secondary end points

a. Number of admissions.
b. General self-efficacy.[37]
c. Healthcare utilisation measured as the number of admissions and unscheduled outpatient visits due to unplanned visits to hospital or primary care centre due to symptoms of COPD or CHF.
d. HADS.[39]
e. HRQoL (EQ-5D).[43]
f. Incremental cost-utility ratios.
g. Comparing HRQoL and costs for healthcare.
h. CAT.[41]
i. The MRC breathlessness scale.[42]
j. SOB-HF.[40]

## Project timeline

Recruitment of participants commenced in August 2017 and was completed in June 2019. During this period, 224 eligible patients had been identified and included in the study. The intervention will continue until early 2020 after which data for the primary outcome will be collected. Data collection for the secondary outcome measures will continue until mid-2021.

## Power calculation

To achieve 80% power based on a p value of 0.05 (two-tailed) for an increase in the proportion of improved patients from 20% to 40% in the primary outcome, a sample size of 91 patients in each group is required. However, we plan to include 110 patients in each group to have some margin for withdrawals/dropouts. Thus, a minimum of 220 patients will be randomised across the two groups.

## Quantitative data analysis

Descriptive statistics will be used to characterise the study groups. Between-group differences will be tested using Fisher's exact test for dichotomous variables and parametric and non-parametric tests will be used for continuous variables. Logistic regression will be used to calculate odds ratios with 95% CIs. Statistical significance will be set to p<0.05 (two-sided). Both intention-to-treat and per-protocol analyses will be conducted. In addition, subgroup analyses will be performed.

## Design of the health economic evaluation

The objective of the health economic analysis is to estimate differences in total societal costs between the two approaches in relation to the difference in quality-adjusted life-years gained (incremental cost-utility ratios) over the 2-year follow-up. This analysis will be followed by analysing its dependence on personal characteristics. The estimation includes two steps: (a) quantify all resource use related to illness and treatment, as well as the HRQoL in physical terms and (b) evaluate uses of resources in monetary terms and calculate a utility measure between minus and plus one from the HRQoL instrument.[44] A ratio will also be created which is the incremental cost (the cost difference between intervention and usual care groups) divided by the incremental effect (corresponding difference in utility).

Resource use will include data from the regional patient register VEGA (held by Region Västra Götaland), which covers all care reimbursed by the region, including hospital, outpatient and primary care conducted within the region but also care received in other regions. Information on prescribed medication and related costs will be obtained from the National Board of Health and Welfare. In addition, data on lost productivity for the patient due to sickness absence (ie, absence from work which can be attributed to sickness) will be retrieved from the MiDAS database (Social Insurance Agency) and self-report in patient questionnaires (since the first 14 days of each sick leave is not covered by official statistics). This information will be used to estimate indirect costs (sometimes called productivity costs) using the human capital approach, which entails multiplying the net days on sick leave by the mean wage and associated social security contributions. The difference in costs and utilities will be compared (assuming that there will be no difference in length of life between the two groups) and cost-utility ratios will be estimated. Several issues are involved in valuing healthcare in monetary terms related to the basic economic–theoretical concept of 'opportunity cost', true societal valuations and marginal costs. We will follow the state-of-the-art practice by making alternative estimates of costs and present the results as sensitivity analyses.[45] Such analyses will be a test of the robustness of the results. HRQoL (measured by EQ-5D) will be collected repeatedly in the patient questionnaires. As a sensitivity analysis, we will use both a general population value set from the UK,[43] as there is no such value set validated for Sweden, and a Swedish experience-based value set[38] for the translation of EQ-5D results into utilities. Sensitivity analyses will also address length of follow-up. Missing data from the questionnaires will be considered as missing at random and thus handled using a multiple imputation strategy.

Dependence of personal characteristics (eg, education and income, collected from the LISA database at Statistics Sweden and through patient questionnaires) will

be tested. The distribution of costs will also be analysed by major stakeholders, that is county councils/regions, market sectors (productivity loss) and individuals/families/friends to facilitate the more limited approaches to economic evaluation. For details, see figure 1.

### Design of the process evaluation

A mixed-method approach will be applied as it is suitable to deal with the complexity of the intervention by combining the strength of qualitative and quantitative methodologies.[46 47] Quantitative and qualitative strands will be integrated by adding findings from different subprojects to understand intervention outcomes and by combining methodologies within subprojects. The primary objective of the process evaluation is to widen and deepen the understanding of intervention mechanisms of impact. Two key questions will guide the evaluation:

► How do people diagnosed with CHF and/or COPD experience that the intervention and their internal and external capabilities influenced intervention outcomes?

► How are HCPs, other invited partners and people diagnosed with CHF and/or COPD interacting the intervention in their everyday life?

### *Exploring experiences of the intervention and internal and external capabilities in relation to programme outcome*

A grounded theory approach[48] will guide data collection and analysis. The method is suitable to study processes and actions and to deepen the understanding of contextual influences. Data will be collected postintervention (6-month follow-up) through individual face-to-face interviews or through phone according to the participant's preferences. Purposeful selection of participants will be used. Initial sampling criteria are set up to reach heterogeneity regarding age, gender, diagnosis, educational level and degree of use of the intervention. Of note, 10–15 interviews are predicted to be needed but inclusion will continue until saturation is reached. A question guide with one opening question and question areas concerning features of intervention content and design and internal/external prerequisites will be used. Interviews will be audio recorded and transcribed verbatim. Initial and focused coding will be used during analysis that is conducted by an experienced qualitative researcher. In the initial coding, each line will be coded close to the data. Later segments of data will be synthesised and explained using a conceptual code (focused coding). Codes within and between interviews will be compared and sorted into categories. Memos will be used to systematically compare codes and to document analytical thoughts.[48]

### *Interactions with intervention in everyday life*

A convergent parallel mixed-method design[49] will be used. User data including which features have been used from the digital platform and ratings of experienced benefits and usability will be collected at the 6-month follow-up

for all participants randomised to the intervention group. These follow-ups will also include open-ended questions connected to the ratings. Self-reported use of time will also be collected for participants and HCPs. In addition, participants in the intervention group will be interviewed individually about experiences of using the intervention. For the participant interviews, purposeful sampling will be applied, sampling criteria set to reach heterogeneity and 10–15 interviews are predicted to be needed. HCPs will be interviewed using focus group methodology to obtain a collective understanding[50] followed by individual in-depth interviews. Focus groups and interviews will be conducted by an experienced researcher. All interviews will be recorded and transcribed verbatim. Descriptive statistics will be applied for quantitative data. Interviews, and answers on open-ended question will be analysed by qualitative content analysis.[51] Qualitative and quantitative data will be analysed separately, thereafter data will be merged into a combined analysis by comparing and matching patterns found in each data source[49] For details, see figure 1.

### Ethics and dissemination

This study, including the consent forms for participants, was approved by the Regional Ethics Board in Gothenburg (Dnr 063-17 and T6013-18). This study conforms with the Declaration of Helsinki.[52] All participants received oral and written informed consent before inclusion in the study.

The consent forms used in this study included information to the participants that their data would be treated confidentially and that the results from the study would be published in scientific journals such that individual persons could not be identified.

### DISCUSSION

This study protocol presents an RCT evaluating the effects of PCC by a combined digital platform and structured telephone support for people with COPD and/or CHF. Previous research has shown that after person-centred telephone support for people with COPD and/or CHF, the level of self-efficacy in the intervention group remained while the control group worsened.[30] However, knowledge about how to promote health in patients with COPD or CHF is limited, so research about alternative ways of promoting health is warranted. This study is unique in an e-health context because it is based on PCC, which is an approach that uses patient resources and goals in the planning of care.[7 9] When providing an e-health intervention for older people (≥65 years), one plausible concern is that they might have problems in understanding and operating a digital platform. To eliminate this potential issue, the HCPs will provide older patients with easy-to-follow instructions developed by a group of specialists in the areas of person-centredness, communication and pedagogics

on how to create and manage their account. The participants can also contact the HCPs if they need help in managing the account.

In addition, the project intends to evaluate the intervention and identify facilitators and barriers during the process. Complex interventions, such as the one described in this protocol, have multiple interacting components. Consequently, the methods and procedures in such interventions are often less standardised.[53] To integrate a process evaluation is therefore highly recommended.[47] A process evaluation provides the opportunity to monitor the intervention and supply information on which parts of the intervention are important and what contextual factors affect the intervention. Evaluation of user experience of the intervention aspires to provide novel knowledge that can be used in clinical practice through an amplified understanding of how the intervention is used and when and for whom it is considered acceptable and beneficial. This novel knowledge can also help to explain programme outcomes.

Interventions must also be cost-effective to be implemented. Accordingly, it is vital to integrate health economic evaluations in the design of a complex intervention. One critique that has been raised against telephone-based interventions is that they may be time-consuming and expensive. However, such interventions have been evaluated and proven cost-effective.[6 54] One example from our previous study is that evaluated PCC using a telephone support programme in which the average time of use was less than 90 min per patient during the 6-month study period.[30]

Conducting an intervention with different or more than one diagnosis can cause difficulties in interpreting the findings. However, this is the reality of today's healthcare system. Many older patients have more than one diagnosis and their care situation is often complex.[55] Both COPD and CHF are common conditions in older people and they share several symptoms (eg, fatigue and breathlessness) and often have similar care issues.[56] In addition, comorbidity is common in COPD and CHF.[57] Thus, including patients with both diagnoses is appropriate when reflecting on the challenges and issues facing the healthcare system today.[30] In open trials, there is always the risk of selection bias. To prevent this, there were strict inclusion and exclusion criteria and all reasons for exclusions of potential participants were recorded.

In conclusion, this RCT, which assesses the effects of PCC by an intervention that combines a digital platform with a structured telephone support programme in patients with COPD and/or CHF, will provide clinically relevant information on the effects of PCC on self-reported symptoms, HRQoL and health economics. Our study will also offer information about the barriers and solutions when applying a digital platform together with structured telephone support based on PCC in patients with COPD or CHF.

**Author affiliations**
¹Institute of Health and Care Sciences, Sahlgrenska Academy, University of Gothenburg, Gothenburg, Sweden
²Centre for Person-Centred Care, University of Gothenburg, Gothenburg, Sweden
³Psychiatric department, Sahlgrenska University Hospital, Gothenburg, Sweden
⁴Närhälsan Research and Development Primary Health Care, Region Västra Götaland, Sweden
⁵Patient representative, The Swedish Heart & Lung Foundation, Stockholm, Sweden
⁶Department of Internal medicine and geriatrics, Sahlgrenska university hospital, Östra, Gothenburg, Sweden
⁷Department of Molecular and Clinical Medicine, Sahlgrenska Academy, University of Gothenburg, Gothenburg, Sweden

**Acknowledgements** We want to thank all the participants for sharing their experiences and the participating primary care centres for support and assistance in conducting the study. We are also grateful to patient research partner Eva Fredholm for constructive discussions throughout the research process.

**Contributors** All authors were involved in the design of the study. LA and SW drafted the manuscript with critical input from EB, AF, HG, EF, KS and IE. IE is the grant holder and project leader. All authors reviewed, edited and approved the final version of the manuscript.

**Funding** This work was supported by The Swedish Heart & Lung Foundation (DNr 20180183), the Swedish Research Council (DNr 2017-01230) and the Centre for Person-Centred Care at the University of Gothenburg (GPCC), Sweden. The funders had no role in analysis or preparation of the manuscript.

**Competing interests** None declared.

**Patient and public involvement** Patients and/or the public were involved in the design, or conduct, or reporting, or dissemination plans of this research. Refer to the Methods section for further details.

**Patient consent for publication** Not required.

**Provenance and peer review** Not commissioned; externally peer reviewed.

**ORCID iDs**
Lilas Ali http://orcid.org/0000-0001-7027-4371
Sara Wallström http://orcid.org/0000-0001-7579-4974
Emmelie Barenfeld http://orcid.org/0000-0002-4945-4623
Hanna Gyllensten http://orcid.org/0000-0001-6890-5162

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
