## [Reviewer comments · BMJ Open]

ARTICLE DETAILS

TITLE (PROVISIONAL)	Person-centred care by a combined digital platform and structured telephone support for people with chronic obstructive pulmonary disease and/or chronic heart failure: Study protocol for the PROTECT randomised controlled trial
AUTHORS	Ali, Lilas; Wallström, Sara; Barenfeld, Emmelie; Fors, Andreas; Fredholm, Eva; Gyllensten, Hanna; Swedberg, Karl; Ekman, Inger

VERSION 1 – REVIEW

REVIEWER	Dr Morag Farquhar University of East Anglia
REVIEW RETURNED	22-Jan-2020

GENERAL COMMENTS	Thank you for the invitation to review this protocol. The following revisions are suggested: Major: 1) Para 101-121 – paragraph needs reworking to present limitations of the e-health first, then the potential advantages/promise. This would present the background research more clearly and logically. At present the two are mixed.2) Line 123 – not sure what is meant by “exacerbated self-efficacy”.3) Line 157 – “not willing to participate” is not an exclusion criterion.4) Line 161 – prognosticating expected survival in COPD is very difficult.5) Line 196-197 – is the health care professional training in the intervention included in the health economic evaluation?6) Line 229-230 – can the people the patient adds or deletes from accessing their account include informal carers (family members or friends who support them in an unpaid role)?7) Line 241 – why are questionnaires being mailed out? Why not use other methods such as email/online/telephone for data collection which would be more in line with the intervention and also easier to return for people with breathlessness or with progressive conditions such as COPD and CHF (avoids having to get to a post box) – some rationale for this choice would be helpful8) Line 253-254 – timing of data collection unclear raising questions about how a composite score can be formed for someone who has died as self-efficacy data can't be collected9) Line 253-254 – how will data be collected for hospitalised patients?10) Line 257 – do the “unscheduled reasons” for admission need to be due to COPD/CHF?11) Line 260 – what is the rationale for having “patient has not been admitted or died” as part of a composite classification of “improved”? – they could increase their self-efficacy but still be admitted for something unrelated (see my Point 10 above)
---

	12) Line 273 – shouldn't the SOB-HF be in the Secondary endpoints list? 13) Section starting at 292 (Health Economic evaluation) – are informal carer costs also captured? If so, how? 14) Line 344-346 – these sentences relating to the qualitative component are very light on detail: when will the patient interviews occur? Where? Who will conduct them? What format (face-to-face? Phone?)? will a topic guide be used? when will the HCP focus groups occur (would have been interesting to do them before, during and after the intervention – is this the case?)? Where? Who will conduct them? What format (by practice or mixed sites?)? will a topic guide be used? 15) Line 346 – what is the proposed sample size of the purposive sample and what is the rationale behind that size? 16) Line 353 – Inadequate qualitative data analysis plan: what method of qualitative analysis will be used? who will do the analysis? Will you involve PPI (given the earlier statements about the importance of involvement in person-centred care)? Also lack of detail on how quant and qual data will be synthesised. 17) Line 374 – think this is the first mention of older people – if this is an important aspect then they should perhaps be referred to earlier 18) Line 376 – how are the instructions being developed? With PPI? Have they been piloted? Minor: 1) Line 63 – should “outcome” be “outcomes”? 2) Line 68 – “patients” needs an apostrophe “patients’ self-care skills” 3) Line 101-102 – there seems to be a word missing in this sentence 4) Line 105-106 – insert “potentially” i.e. “it is potentially accessible everywhere” (as it relies on their being an Internet connection) 5) Line 106-107 – I am not following how “traditional treatment” can be delivered via the Internet 6) Line 107 – suggest delete “dare or want to” as not enough information given to explain these points and the sentence still makes sense 7) Line 108 – change “to have contact” to “of having contact” 8) Line 111-113 – sentence beginning “People yearn” contradicts the previous sentence beginning “Many people” when they are presented separately – would be better to connect them with the term “yet” – then they flow. 9) Line 114-115 – refers to “previous research” but no reference given 10) Line 116 – “has” should be “have” 11) Line 116-117 – part sentence beginning “it is an anchorage...” – not clear what this part sentence means – may need rewording 12) Line 151 – reorder words in sentence to: “Study participants that fulfil the inclusion criteria of the study will be recruited...” 13) Line 248 – capitals needed on Hospital Anxiety and Depression Scale 14) Line 249 – capitals needed on Shortness of Breath in Heart Failure 15) Line 255 – “Patient is classified...” should be “Patient will be classified...” 16) Line 284 – change “randomised to the two groups” to “randomised across the two groups” 17) Line 285 – heading should be “Quantitative Data Analysis” – as it only covers this 18) Line 335 – think there may be something missing or a punctuation error in this question – should it rather be: “How is the
--	--

	intervention content, designed to influence intervention outcomes, experienced by..."
--	---

REVIEWER	Dr Carolyn Deighan NHS Lothian, Scotland, UK Carolyn Deighan is employed at the Heart Manual Department, NHS Lothian, Scotland.
REVIEW RETURNED	23-Jan-2020

GENERAL COMMENTS	Thank you for inviting me to review this protocol. The authors highlight the pressing need for interventions for both COPD and CHF to optimise care to support the management of these conditions. I note that the recruitment has already commenced in August 2017. Since this time there may have been investigations that have used patient centred care through e-health services – does the statement that claims that this has not been previously investigated in the abstract refer to Gothenburg or further afield e.g., the UK? At what time point was the literature search for such studies carried out and for what years? How was Person Centred Care defined in the search? Background Person centred care (PCC) is broadly described conveying a sense of the ethos of PCC and there are references studies showing positive effects using such an approach. However, for the purpose of clarity in this protocol PCC could be better operationalised. Key components of PCC used in the intervention could be more clearly outlined and presented. A schematic diagram would be welcome here. What is the occupation of the healthcare professionals – and what training was provided? E-Health section Some of the English language needs to be corrected. Some sentences do not make sense and are grammatically incorrect: lines 104, 105, 116. Please provide a reference to back up the statement on line 114 & 115. Line 123: What is exacerbated self-efficacy? Enrolment and randomisation Again a diagram would be welcome to capture the process of above and also display the intervention and data collection schedule . Digital platform It appears that the platform has already been developed with the input of health professionals. However, there is little information about the digital platform or its development process. If this has been developed, a table with the headings would be useful to show the content and functionality of the platform. Where is the platform hosted? Is it a mobile platform? Health economic evaluation This section would also benefit from a diagram to capture the schedule. Design of the process evaluation Again this could be more concisely presented with the aid of a flow chart. In summary, there are parts that can be better described and defined to allow for a thoroughly clear protocol.
--

REVIEWER	Palmira Bernocchi ICS MAUGERI, ITALY
REVIEW RETURNED	28-Jan-2020

GENERAL COMMENTS	The authors report on the study design and methodology of a randomized controlled trial evaluating the effect of a person center care combined by a digital platform and structured telephone support in patients with COPD and / or chronic heart failure. The intervention group is compared to a control group receiving usual care. The authors present a nicely planned study protocol and discussed the methodology. The study seems well planned, with sound primary and secondary outcomes. It is a very nicely written and well-prepared manuscript, which is certainly of high interest to the readership. The study for primary outcome is ended. The data collection for secondary outcome will continue until 2021. The topic is of interest. The manuscript is well written. As other partially open trials, there are some other remaining risks of bias, such as selection, performance and attrition. Please, provide arguments to prevent such biases. The inclusion criteria is "diagnosed with COPD or CHF" without any level of intensity of disease: GOLD class and NYHA class. Are all severity classes included? The authors should better specify as the control group is followed. Would it be interesting to know who are the health professionals who make up the HCPs? Who makes the calls? what are the duties of these personnel? Can patients also call HCPs in case of worsening in sign and symptoms? How are managed sign and symptoms during structured phone call, if there were? Counselling and educational support for adherence to therapy and physical activity? Power calculation On what basis the sample size was calculated. Based on the data you previously published on in patients followed with PCC and structured telephone support available literature. Please specify?
--

VERSION 1 – AUTHOR RESPONSE

Reviewer nr 1	Remarks
Thank you for the invitation to review this protocol. The following revisions are suggested:	Thank you very much for reviewing our manuscript and for all your comments.
Major: Para 101-121 – paragraph needs reworking to present limitations of the e-health first, then the potential advantages/promise. This would present the background research more clearly and logically. At present the two are mixed.	Thank you for your comment. We have restructured the eHealth section in the background of our manuscript.
2) Line 123 – not sure what is meant by “exacerbated self-efficacy”.	1) We hope that we have now clarified this sentence.
3) Line 157 – “not willing to participate” is not an exclusion criterion.	2) Thank you for this remark! We have now edited this.
4) Line 161 – prognosticating	3) We agree that it is difficult, but this assessment was

expected survival in COPD is very difficult.	made by experienced clinical physicians.
5) Line 196-197 – is the health care professional training in the intervention included in the health economic evaluation?	4) Thank you for this remark. The health economic evaluations is planned to include calculations that considers this cost.
6) Line 229-230 – can the people the patient adds or deletes from accessing their account include informal carers (family members or friends who support them in an unpaid role)?	5) Yes, the patients can give access to their family members or friends who support them and they can customize what they can access. Such as for e.g. just being able to access the health plan and messages. We have now clarified this in the manuscript. People invited by the patient can themselves not invite others to the patient's account.
7) Line 241 – why are questionnaires being mailed out? Why not use other methods such as email/online/telephone for data collection which would be more in line with the intervention and also easier to return for people with breathlessness or with progressive conditions such as COPD and CHF (avoids having to get to a post box) – some rationale for this choice would be helpful	Thank you for the remark. Paper questionnaires were chosen since not all of the instruments are validated for online use. We have added this explanation to the manuscript.
8) Line 253-254 – timing of data collection unclear raising questions about how a composite score can be formed for someone who has died as self-efficacy data can't be collected	Thank you for the comment. All participants that die are automatically classified as deteriorated so information on self-efficacy is not needed. We have clarified this in the manuscript.
9) Line 253-254 – how will data be collected for hospitalised patients?	Thank you for the comment. All questionnaires are sent to the patients' homes. If the patient is readmitted, he/she is asked to answer the questionnaires ones discharged.
10) Line 257 – do the “unscheduled reasons” for admission need to be due to COPD/CHF?	Yes, this has been clarified in the manuscript.
11) Line 260 – what is the rationale for having “patient has not been admitted or died” as part of a composite classification of “improved”? – they could increase their self-efficacy but still be admitted for something unrelated (see my Point 10 above)	We have considered admittance due to COPD/CHF as such a large burden (both for society and the patient), i.e. an adverse event that we think overrides an improvement in self-efficacy. Admittance for something unrelated will not have an impact on the composite classification.
12) Line 273 – shouldn't the SOB-HF be in the Secondary endpoints list?	Thank you for this keen observation. We have added the SOB to the list.
13) Section starting at 292 (Health Economic evaluation) – are informal carer costs also captured? If so, how?	No, unfortunately we have no good way of capturing informal carer costs.
14) Line 344-346 – these sentences relating to the qualitative component are very light on detail: when will the	Thank you for your comments. We have now clarified these questions in the manuscript and they have been extensively added to the section of design of the

patient interviews occur? Where? Who will conduct them? What format (face-to-face? Phone?)? will a topic guide be used? when will the HCP focus groups occur (would have been interesting to do them before, during and after the intervention – is this the case?)? Where? Who will conduct them? What format (by practice or mixed sites?)? will a topic guide be used?	process evaluation.
15) Line 346 – what is the proposed sample size of the purposive sample and what is the rationale behind that size?	Thank you for this comment. We have now added the sample size.
16) Line 353 – Inadequate qualitative data analysis plan: what method of qualitative analysis will be used? who will do the analysis? Will you involve PPI (given the earlier statements about the importance of involvement in person-centred care)? Also lack of detail on how quant and qual data will be synthesised.	Thank you for these comments and questions, we have added the requested information regarding methods, qualitative analysis plans and what kind of a mixed-methods approach that will be applied. Our patient research partner will according to our current plan not be involved in the analysis of data, but in other parts of the research process such as co-creating question guides, and having dialogues about how to present and disseminate findings to reach the targeted audience.
17) Line 374 – think this is the first mention of older people – if this is an important aspect then they should perhaps be referred to earlier	Thank you for this observation, we have now added this early in the background of the manuscript.
18) Line 376 – how are the instructions being developed? With PPI? Have they been piloted?	Thank you for the comment. The instructions were developed by a group of specialists in the areas of person-centredness, communication and pedagogics. This clarification has been added to the manuscript.
Minor:	
1) Line 63 – should “outcome” be “outcomes”?	Yes, edited.
2) Line 68 – “patients” needs an apostrophe “patients’ self-care skills”	Yes, edited.
3) Line 101-102 – there seems to be a word missing in this sentence	Thank you for this observation. We have now edited this.
4) Line 105-106 – insert “potentially” i.e. “it is potentially accessible everywhere” (as it relies on their being an Internet connection)	Yes, edited.
5) Line 106-107 – I am not following how “traditional treatment” can be delivered via the Internet	Thank you for the comment. We have changed the sentence.
6) Line 107 – suggest delete “dare or want to” as not enough information given to explain these points and the sentence still makes sense	Yes, edited.

7) Line 108 – change “to have contact” to “of having contact”	Yes, edited.
8) Line 111-113 – sentence beginning “People year” contradicts the previous sentence beginning “Many people” when they are presented separately – would be better to connect them with the term “yet” – then they flow.	Yes, edited.
9) Line 114-115 – refers to “previous research” but no reference given	Thank you, a reference has been added.
10) Line 116 – “has” should be “have”	Yes, edited.
11) Line 116-117 – part sentence beginning “it is an anchorage...” – not clear what this part sentence means – may need rewording	Thank you for the comment. This sentence has been removed.
12) Line 151 – reorder words in sentence to: “Study participants that fulfil the inclusion criteria of the study will be recruited...”	Yes, edited.
13) Line 248 – capitals needed on Hospital Anxiety and Depression Scale	Yes, edited.
14) Line 249 – capitals needed on Shortness of Breath in Heart Failure	Yes, edited.
15) Line 255 – “Patient is classified...” should be “Patient will be classified...”	Yes, edited.
16) Line 284 – change “randomised to the two groups” to “randomised across the two groups”	Yes, edited.
17) Line 285 – heading should be “Quantitative Data Analysis” – as it only covers this	Yes, edited.
18) Line 335 – think there may be something missing or a punctuation error in this question – should it rather be: “How is the intervention content, designed to influence intervention outcomes, experienced by...”	Yes, edited.
Reviewer nr 2	Remarks
Thank you for inviting me to review this protocol. The authors highlight the pressing need for interventions for both COPD and CHF to optimise care to support the	Thank you for reviewing this protocol. Searches for studies conducted on Person-centred care by a combined digital platform and structured telephone support for people with chronic obstructive pulmonary disease and/or chronic heart failure were conducted

management of these conditions. I note that the recruitment has already commenced in August 2017. Since this time there may have been investigations that have used patient centred care through e-health services – does the statement that claims that this has not been previously investigated in the abstract refer to Gothenburg or further afield e.g., the UK? At what time point was the literature search for such studies carried out and for what years? How was Person Centred Care defined in the search?	both before study start and in connection with writing this protocol. At neither of these occasions could we find other studies using a person-centred approach grounded on the philosophy of personhood as an approach through an eHealth service such as a digital platform. We search both for studies at our centre and in other places.
Background Person centred care (PCC) is broadly described conveying a sense of the ethos of PCC and there are references studies showing positive effects using such an approach. However, for the purpose of clarity in this protocol PCC could be better operationalised.	Thank you for the comment. We have added examples of operationalization in the background.
Key components of PCC used in the intervention could be more clearly outlined and presented. A schematic diagram would be welcome here.	Thank you for this comment. We have now clarified this in the protocol and added a figure to outline the process of the intervention/design.
What is the occupation of the healthcare professionals – and what training was provided?	Thank you for the comment. We have added this information to the manuscript.
E-Health section Some of the English language needs to be corrected. Some sentences do not make sense and are grammatically incorrect: lines 104, 105, 116. Please provide a reference to back up the statement on line 114 & 115.	Thank you for the comment. We have edited the manuscript. This statement is backed up by ref 26.
Line 123: What is exacerbated self-efficacy?	Thank you for the remark. The sentence has been re-worded.
Enrolment and randomisation Again a diagram would be welcome to capture the process of above and also display the intervention and data collection schedule .	Thank you for this comment. We have now added a figure to outline the process of the intervention/design.
Digital platform It appears that the platform has already been developed with the input of health professionals. However, there is little information about the digital platform or its development process.	Thank you for the comment. We have added information on the development process and input from HCPs and patient representatives.

If this has been developed, a table with the headings would be useful to show the content and functionality of the platform.	Thank you. Information about the functionalities of the platform has been added.
Where is the platform hosted? Is it a mobile platform?	The platform is hosted by the University of Gothenburg. It is a digital platform but not an app for mobile phones.
Health economic evaluation This section would also benefit from a diagram to capture the schedule.	Thank you. We have added a figure that includes this information.
Design of the process evaluation Again this could be more concisely presented with the aid of a flow chart.	Thank you. We have added a figure that includes this information.
In summary, there are parts that can be better described and defined to allow for a thoroughly clear protocol.	Thank you for your comments. Hopefully the protocol is clearer now.
Review nr 3	
The authors report on the study design and methodology of a randomized controlled trial evaluating the effect of a person center care combined by a digital platform and structured telephone support in patients with COPD and / or chronic heart failure. The intervention group is compared to a control group receiving usual care. The authors present a nicely planned study protocol and discussed the methodology. The study seems well planned, with sound primary and secondary outcomes. It is a very nicely written and well-prepared manuscript, which is certainly of high interest to the readership.	Thank you so much for your review of our protocol and for those kind words.
The study for primary outcome is ended. The data collection for secondary outcome will continue until 2021. The topic is of interest. The manuscript is well written.	Thank you very much for your comments. The primary outcome has now ended but had not at time of submission of this manuscript.
As other partially open trials, there are some other remaining risks of bias, such as selection, performance and attrition. Please, provide arguments to prevent such biases.	Thank you for this comment. We have now added arguments concerning bias in the manuscript.
The inclusion criteria is "diagnosed with COPD or CHF" without any level of intensity of disease: GOLD class and NYHA class. Are all severity classes included?	Yes, all classes are included in this study as this reflects the primary care population.

The authors should better specify as the control group is followed.	Thank you for the comment. This information has now been added to the manuscript and in the added figure.
Would it be interesting to know who are the health professionals who make up the HCPs? Who makes the calls? what are the duties of these personnel?	Thank you. This has been clarified in the manuscript.
Can patients also call HCPs in case of worsening in sign and symptoms?	Yes, they are free to contact the HCPs about any concerns.
How are managed sign and symptoms during structured phone call, if there were?	Thank you for the comment. There were no mandatory follow-up of symptoms during the conversations but they discussed if the patient wanted to.
Counselling and educational support for adherence to therapy and physical activity?	Thank you for the comment. There were no mandatory counselling and educational support for adherence to therapy and physical activity but this was discussed if the patient wanted to.
Power calculation On what basis the sample size was calculated. Based on the data you previously published on in patients followed with PCC and structured telephone support available literature. Please specify?	Thank you for the comment. The power calculation and sample size is based on the following studies:  • Fors A, Ekman I, Taft C, et al. Person-centred care after acute coronary syndrome, from hospital to primary care—a randomised controlled trial. International journal of cardiology 2015;187:693-99. • Fors A, Blanck E, Ali L, et al. Effects of a person-centred telephone-support in patients with chronic obstructive pulmonary disease and/or chronic heart failure—A randomized controlled trial. PloS one 2018;13(8):e0203031.

VERSION 2 – REVIEW

REVIEWER	Dr Morag Farquhar University of East Anglia (UEA), UK
REVIEW RETURNED	04-Mar-2020
GENERAL COMMENTS	The protocol is much improved by the inclusion of greater detail on the qualitative component. Minor revisions required: 1) The revision includes reference to "Figure 1" but there was no figure included to review (that I could see) 2) The English language needs some attention in some of the newly added segments, but I imagine that the editorial team will assist with this?
REVIEWER	Dr Carolyn Deighan NHS Lothian
REVIEW RETURNED	13-May-2020
GENERAL COMMENTS	My comments have been addressed in this revised version.

	Thank you.
REVIEWER	Palmira Bernocchi Italy
REVIEW RETURNED	12-Mar-2020
GENERAL COMMENTS	Thank you for inviting me to review again this protocol. The authors responded satisfactorily to the reviewer's requests.